# Prevalence and Predictive Factors of Masked Depression and Anxiety among Jordanian and Palestinian Couples: A Cross-Sectional Study

**DOI:** 10.3390/healthcare10091679

**Published:** 2022-09-02

**Authors:** Deema Jaber, Haneen A. Basheer, Lina Elsalem, Mohammad Dweib, Maysa Khadra, Rami Abduljabbar, Rawan Ghazwi, Hamza Alhamad

**Affiliations:** 1Department of Clinical Pharmacy, Faculty of Pharmacy, Zarqa University, Zarqa 13110, Jordan; 2Department of Pharmacology, Faculty of Medicine, Jordan University of Science and Technology, Irbid 22110, Jordan; 3Department of Clinical Pharmacy, Faculty of Pharmacy, Hebron University, Hebron P720, Palestine; 4Department of Obstetrics and Gynecology, Faculty of Medicine, University of Jordan, Amman 11942, Jordan; 5Department of Clinical Pharmacy, Faculty of Pharmacy, University of Jordan, Amman 11942, Jordan

**Keywords:** psychology, prevalence, fertile couples’ masked anxiety, fertile couples’ masked depression

## Abstract

Although anxiety and depression are among the most prevalent mental disorders worldwide, they continue to gain less attention than their physical counterparts in terms of health care provision and population mentalisation. This cross-sectional study explores and compares the national prevalence of depression and anxiety signs/symptoms and well as identifying associated socio-demographic factors among Jordanian and Palestinian fertile couples. Four-hundred and sixty-nine participants were eligible for inclusion and agreed to participate in the study. The mean score for HAM-A and BDI-II were 12.3 ± 8.2 and 15.30 ± 10.0, respectively. According to the grading of HAM-A and BDI-II, the majority of the participants have graded themselves to be mildly anxious (*N* = 323, 68.9%) and around one third of participants (*N* = 148, 31.6%) moderately to severe depressed. The suicidal intent was remarkable and of concern where around 18.6% of participants had suicidal thoughts and wishes. There was a significant correlation between both HAM-score and BDI-II score and age [*p* = 0.01, *p* = 0.011, respectively], body weight [*p* = 0.01, *p* = 0.006, respectively], and total monthly income [*p* < 0.001, *p* < 0.001, respectively]. Our findings ought to alert healthcare professionals and other interested parties that there is a high burden of anxiety and depression symptoms among Jordanian and Palestinian couples. To support Jordanian and Palestinian couples’ mental health, healthcare professionals, researchers, and educators favoured to concentrate on creating efficient and culturally relevant education, preventive, and intervention procedures utilising evidence-based guidelines.

## 1. Introduction

Anxiety and depression are among the most prevalent mental disorders worldwide, and they seem to continually increase in incidence and prevalence [1,2,3]. Such diseases have been associated with increased medical morbidity and mortality, functional impairment and diminished quality of life [4,5]. In addition, they have significant negative impacts on public health, physical conditions and economics [6]. Information about their prevalence and correlation is a key element for prevention and management [4,7]. The association between depression and anxiety is particularly strong, and quality of life impairments due to anxiety are comparable to the magnitude of dysfunction from depression [6]. Worldwide studies have confirmed the association between anxiety and depression with many sociodemographic factors including gender, age, ethnicity, income, environmental and social factors [8].

Although the importance of anxiety and depression was recently acknowledged in the World Health Organisation (WHO) report on disability, they continue to gain less attention than their physical counterparts in terms of health care provision and population mentalisation [9,10]. Researchers encourage awareness of mental disorders and diagnose patients outside of mental health facilities in order to obtain the necessary precision for prevalence studies. This is especially true in Arab nations, where psychological issues are not seen as requiring professional assistance, and those who do seek assistance frequently do not speak for the general population because they are typically more severely afflicted and suffer from comorbid disorders [11,12]. These patients are at increased risk of suicide, cardiovascular-related events and death [13,14]. Age, gender, marital status, education level, and economic status have all been repeatedly identified as sociodemographic variables that have a significant impact on explaining the variation in the prevalence of depression and anxiety worldwide [15,16].

There is currently a significant disparity between educational options and employment prospects in Arab countries. Although the Arab population’s literacy and education levels have dramatically increased, there are still few job prospects available, which leaves record numbers of young people unemployed once they complete their studies [17,18].

Various screening tools are used to detect anxiety and depression, including the Hamilton Rating Scale for Anxiety (HAM-A) [19] and Beck’s Depression Inventory-II (BDI-II) [20]. Patients with positive results on any screening tool may need referrals to mental health professionals to assess whether they meet the criteria in the Diagnostic and Statistical Manual of Mental Disorders, Fifth Edition (DSM-5) [21]. Most patients with anxiety and depression can be diagnosed by a primary care physician.

Population surveys have explored the prevalence of anxiety and depression in Western countries [1,2,3,4,5], while data from developing countries, such as those in the Middle East, are comparatively scarce. It is challenging to develop, put into practice, and disseminate effective interventions to enhance the prevention, diagnosis, and treatment of the population’s depression and anxiety in the region due to the paucity of research on the current profile in the Arab nations generally and Jordan/Palestine in particular. Lack of awareness programmes about anxiety and depression, unavailability of appropriate therapies, competing clinical demands, social issues, the lack of patient acceptance of the diagnosis and high health care costs, especially in the absence of health insurance, have been thought to be among the most important barriers to the identification, diagnosis and treatment of patients in many developing countries, such as Jordan and Palestine [22,23,24,25]. According to recent meta-analysis reviews, there was a higher burden of anxiety and depression in Africa and the Middle East compared to other areas of the world [26,27].

To the best of the authors’ knowledge, this is the first study that explores and compares national prevalence of depression and anxiety signs and symptoms among Jordanian and Palestinian fertile couples, as well as identifying associated sociodemographic characteristics. To address the current gap in the literature, this study aims to explore anxiety and depression in relatively healthy young to middle-aged couples. The outcomes of this study provide important and valuable information that can be significantly useful for future research on mental health improvement programs among Jordanian and Palestinian populations, as well as the development of mental health policies.

## 2. Materials and Methods

### 2.1. Study Design

This cross-sectional study was conducted in Jordan and Palestine from December 2018 to February 2019. The study population included married couples living in various areas of Jordan and Palestine. Both Palestinian and Jordanian societies are regarded as being in their formative years, and a sizable portion of the populace is regarded as being of reproductive age. The major objective of this research project, which consists of various phases, is to examine this population’s childbearing age group. Accordingly, participants were included in the study if they were married couples, Arabic speaking and willing to participate in the study.

During the study period, a convenience sample of participants was approached to voluntarily and anonymously participate in this study. Several health centres, the community, and some clinics that serve us, such as the family medicine and gynaecology-obstetrics clinics of the aforementioned hospitals, were used to compile the research sample.

### 2.2. Data Collection

Data were collected using a structured interviewer-administered questionnaire. Trained clinical pharmacists interviewed the identified couples. Each participant was interviewed separately from his/her partner to decrease the risk of influencing responses.

Signed informed consent was obtained from all couples before data collection and after explaining the details of the research, its procedures, the potential benefits and answering their questions satisfactorily.

### 2.3. Sample Size Calculation

The sample size was calculated based on the number of study couples required to perform regression analysis (5–20 study participants per predictor) [28]. Using a medium number of 10 study participants per predictor, because we have 20 predictors, a minimum sample size of 200 was considered representative for each of the two countries (Jordan and Palestine). Therefore, we needed a sample size of 400 to be considered representative.

### 2.4. Clinical Measures

The validated Arabic versions of Beck’s Depression Inventory-II (BDI-II) [29] and Hamilton Rating Scale for Anxiety (HAM-A) [30] screening tools were used. The HAM-A is the most widely used semi-structured assessment scale in treatment outcome studies of anxiety. It is a 14-item test measuring the severity of anxiety symptoms and is designed for both psychological and somatic symptoms. It includes questions meant to measure anxious mood; tension (startle response, fatigability, restlessness); fears (including of the dark/strangers/crowds); insomnia; cognitive symptoms (poor memory/difficulty concentrating); depressed mood (including anhedonia); somatic symptoms (aches and pains, stiffness, bruxism); sensory symptoms (tinnitus, blurred vision); cardiovascular symptoms (tachycardia and palpitations); respiratory symptoms (chest tightness, choking); gastrointestinal symptoms (such as those experienced with irritable bowel syndrome); genitourinary symptoms (urinary frequency, loss of libido); autonomic symptoms (dry mouth, tension headache) and observed behaviour at interview (restlessness, fidgeting). For the 14 items, the values on the scale range from zero to four, where zero means there is no anxiety, one indicates mild anxiety, two indicates moderate anxiety, three indicate severe anxiety and four indicates very severe or grossly disabling anxiety. The total anxiety score ranges from zero to 56. The seven psychic anxiety items reveal a psychic anxiety score ranging from zero to 28. The remaining seven items yield a somatic anxiety score that also ranges from zero to 28. Scores in the 0–17 range may be indicative of mild anxiety levels, scores in the 18–24 range are indicative of mild to moderate anxiety levels, scores in the 25–30 range are indicative of moderate to severe anxiety levels and scores in the 31–56 range are indicative of very severe anxiety levels. The BDI-II is composed of items relating to symptoms of depression, such as hopelessness and irritability; cognitions, such as guilt or feelings of being punished; and physical symptoms, such as fatigue, weight loss and lack of interest in sex. The BDI-II has 21 questions, and each answer is scored on a scale of 0 to 3. The highest possible total score for the whole test would be 63, and the lowest possible score for the test would be zero. Scores in the 0–10 range are considered normal, scores in the 11–16 range are indicative of mild mood disturbance levels, scores in the 17–20 range are indicative of borderline clinical depression levels, scores in the 21–30 range are indicative of moderate depression levels, scores in the 31–40 range are indicative of severe depression levels and scores over 40 are indicative of extreme depression levels. The BDI-II has good internal consistency (α  =  0.92) and 1-week test–retest reliability (r  =  0.93) [20]. The HAM-A exhibited good construct validity, showing statistically significant relationships with independent self-report measures of generalised anxiety and other anxiety variables [31].

### 2.5. Ethical Considerations

The World Medical Association Declaration of Helsinki guidance was followed in the study which was approved by the Deanship of Research and the Institutional Review Board committee of Zarqa University (ZU), JUH and KAUH (IRB number 64/118/2018) to ensure ethical procedures in data collection and analysis.

### 2.6. Statistical Analysis

Following data collection, the questionnaire responses were coded and entered into a customised database using the Statistical Package for the Social Sciences (SPSS), Version 23.0 (IBM Corp., Armonk, NY, USA). The descriptive data were analysed using frequency/percentage for qualitative data and mean/standard deviation for quantitative data. Independent t-tests and ANOVA were used to test the assessment of the relationship between different demographic variables and BDI-II and HAM-A scores. Independent factors correlating with BDI-II and HAM-A scores were determined using simple linear regression. Subsequently, any variable with a *p*-value of < 0.05 was entered into multiple regression analysis utilising backward elimination method. Statistical significance was defined as *p* < 0.05. Checks for normality were carried out using the Shapiro–Wilk test (with *p*-value ≥ 0.05 indicates a normally distributed continuous variable). Before performing the multiple linear regression analysis, it was made sure that there was no multicollinearity between the independent variables; variance inflation factor (VIF) values were <10. Variables were selected after checking their independence where tolerance values > 0.1. Homoscedasticity assumptions were checked using the Breusch–Pagan test, with a *p*-value ≥ of 0.05 indicating the absence of heteroscedasticity. The Pearson product–moment correlation coefficient was computed to assess the relationship between the BDI-II and HAM-A scores and different participants’ continuous demographic data, such as age, weight, length of marriage, total monthly income and total HAM-A score.

## 3. Results

### 3.1. Demographic Characteristics

Five-hundred and ten couples were approached from the general population. Four-hundred and sixty-nine participants were eligible for inclusion and agreed to participate in the study (response rate 91.9%). The socio-demographic characteristics of study participants are presented in Table 1. Around two thirds of participants were Jordanians [60.8% (*N* = 285)] with a mean age of 36 years (±9.70). More than two thirds of respondents were well educated (having and BSc education or higher) [63.1% (*N* = 296)], closely followed by high school qualification or less [36.4% (*N* = 171)]. More than half of participants [52.0% (*N* = 244)] were from nuclear families with an average period of marriage of 10.5 ± 7.9 years. Most of the respondents [72.1% (*N* = 338)] were working with an average total family monthly income of 827 ± 655 Jordanian dinar; [78.5% (*N* = 368)] were medically insured. More than one third of respondents were active smokers including those with regular pipe (shisha) smoking [32.2% (*N* = 151)]. The majority of participants believe that their health was excellent, [30.1% (*N* = 141)] or at least very good [47.1% (*N* = 221)].

### 3.2. Depression and Anxiety Characteristics

The mean score for HAM-A and BDI-II were 12.3 ± 8.2 and 15.30 ± 10.0, respectively. According to the grading of HAM-A and BDI-II, the majority of the participants have graded themselves to be mildly anxious (*N* = 323, 68.9%) and around one third of participants (*N* = 148, 31.6%) moderately to severe depressed. Symptoms of depression, defined by a BDI-II > 17, were seen among 219 participants (46.8%); a BDI-II > 20 was observed among 153 participants (32.7%). These 153 participants (32.7%) met the criteria for major depression (BDI-II > 20) at the time of the study and clinically they need to consult the psychiatrist consultation. The suicidal intent was remarkable and of concern where around 18.6% of participants had suicidal thoughts and wishes. The summary of BDI-II and HAM-A scores and grading of study participants is presented in Table 2.

### 3.3. Associated Related Factors of Anxiety and Depression among Participants

#### 3.3.1. Correlation between BDI-II Score and Age, Period of Marriage, Total Monthly Income and Total HAM-A Score

A Pearson product–moment correlation coefficient was computed to assess the relationship between BDI-II score and different participants’ demographic variables such as age, weight, period of marriage, total monthly income and total HAM-A score. There was a significant correlation between BDI-II score and age [r = −0.124, *N* = 423, *p* = 0.011], BDI-II score and weight [r = −0.129, *N* = 453, *p* = 0.006], BDI-II score and total monthly income [r = −0.252, *N* = 476, *p* < 0.001], BDI-II and to total HAM-A score [r = 0.721, *N* = 468, *p* < 0.001]. On the contrary, no significant correlation was found between BDI-II score and period of marriage [r = −0.017, *N* = 464, *p* = 0.71]. A significant inverse relationship was found between BDI-II score and age, weight and total monthly income. A significant positive relationship was found between BDI-II score and anxiety total HAM-A score.

#### 3.3.2. Correlation between HAM-A Score and Age, Period of Marriage, Total Monthly Income and Total BDI-II Score

A Pearson product–moment correlation coefficient was computed to assess the relationship between HAM-A score and different participants’ continuous demographic variables such as age, weight, period of marriage, total monthly income and total BDI-II score. There was a significant correlation between HAM-A score and age [*r* = −0.126, *N* = 423, *p* = 0.01], HAM-A score and weight [*r* = −0.121, *N* = 453, *p* = 0.01], HAM-A score and total monthly income [*r* = −0.219, *N* = 476, *p* < 0.001], HAM and to total BDI-II score [*r* = + 0.721, *N* = 468, *p* < 0.001]. On the contrary, no significant correlation was found between HAM-A score and period of marriage [*r* = −0.008, *N* = 464, *p* = 0.87]. A significant inverse relationship was found between BDI-II score and age and total monthly income. A significant positive relationship was found between BDI-II score and anxiety total HAM-A score.

#### 3.3.3. Correlation between BDI-II and HAM-A Scores and Categorical Participants’ Demographic Variables

Univariate analysis was conducted to assess the related factors associated with both HAM-A and BDI-II scores. Results show that five variables including; gender, family type, education level, employment and participants’ health evaluation, were significant variables of both HAM-A and BDI-II scores (*p* < 0.05) mainly among Jordanian population. Data are summarised in Table 3.

As detailed earlier and regarding Jordanian couples, all variables with *p* < 0.05 in the single predictor analysis were entered into multiple linear regression to identify the significant and independent predictors for both HAM-A and BDI-II scores. Data are summarised in Table 4.

The HAM-A score was found to be significantly and independently correlated with four different variables (*p* < 0.05). Standardised beta values, which illustrate the separate contribution of HAM-A score, revealed that health evaluation (*B* = 0.226; *p* < 0.001) had the greatest contribution of unique variance followed by gender (*B* = 0.170; *p* = 0.015), family type (*B* = 0.141; *p* = 0.024), and education level (*B* = −0.125; *p* = 0.047) contributed significantly to the model. The positive *B* values indicate that higher HAM-A score was significantly and independently correlated with three factors in the final model; participants’ health evaluation, gender and family type. However, the negative *B* value of education level indicated that this variable in the final model was associated with a lower HAM-A score.

The BDI-II score was found to be significantly and independently correlated with three different variables (*p* < 0.05). Standardised beta values, which illustrate the separate contribution of BDI-II score, revealed that health evaluation (*B* = 0.250; *p* < 0.001) had the greatest contribution of unique variance followed by working status (*B* = 0.139; *p* = 0.040), and education level (*B* = −0.166; *p* = 0.007) contributed significantly to the model. The positive *B* values indicate that higher BDI-II score was significantly and independently correlated with three factors in the final model; participants’ health evaluation and working status. However, the negative *B* value of education level indicated that this variable in the final model was associated with a lower BDI-II score.

## 4. Discussion

The methods used in Jordan and Palestine for the diagnosis, treatment, and prevention of many mental illnesses fall far short of the global recommendations. To our knowledge, this is the first study to explore and compare national prevalence of depression and anxiety signs and symptoms as well as identify associated socio-demographic factors among Jordanian and Palestinian fertile couples in the child-bearing age.

The study reveals that prevalence rate for moderate to severe or higher depression scores was 32.7%, with 30.3% of the participants reporting mild–moderate or higher anxiety scores. However, it is crucial to remember that this is just a rough approximation.

The clinical picture of depression and anxiety in young couples can be mimicked or confused by clinical presentations for numerous medical and mental illnesses, as well as responses to psychosocial stressors [32,33,34]. Psychiatrists must therefore conduct diagnostic interviews as part of their examination for potential differential diagnoses. It may be challenging to compare our results to estimates from the Western literature because these studies frequently give prevalence rates based on diagnostic tests rather than self-reported questionnaires. However, it is crucial to understand how our prevalence rate would stack up against the global prevalence.

Our survey reveals that the intensity of depression and anxiety is higher in women, with lower family monthly incomes, who live in extended families, with lower education levels, who are unemployed, and those who rate their own health less favorably. Clinically, it is crucial to recognise this group of sociodemographic traits linked to a higher risk of depression in Jordanian and Palestinian fertile couples because it can be used as a starting point to identify patients who may need additional help and referrals to mental health professionals.

Epidemiological data worldwide including the Arab world indicate that both depression and anxiety are more prevalent in women than in men [35,36,37]. Although the exact processes underlying this trend remain unclear, this could be primarily result of gender-related hormonal profiles and depend less on race, culture, diet, education and numerous other potentially confounding social and economic factors [35,36,37,38,39]. At this time, it is likely that unpleasant childhood experiences, childhood and adolescent depression and anxiety disorders, societal roles associated with unfavourable experiences, and psychological characteristics linked to susceptibility to life events and coping mechanisms will be implicated. However, it is still difficult to integrate the factors influencing gender differences in mental problems into comprehensive etiological models [40,41].

The average family monthly income of our participants is 827 JOD. According to the most recent data made public by the department of economic statistics at the Jordanian Department for Statistics (2010) [42] indicates that the poverty line in Jordan has climbed significantly over the past few years, reaching 800 JOD. As a result, fertile couples in this study can be regarded as being at the poverty line. Several studies have shown that unemployment increases the risk for depression and anxiety [43,44,45]. Economically, Jordan and Palestine are low-income countries with difficult economic conditions in which Jordanians and Palestinians have been enduring a very serious 10-year recession and have an unemployment rate of over 20% [18]. Moreover, there is evidence suggesting that the increased prevalence of various mental disorders is the result of the recent economic and financial crisis and high unemployment rates [46]. It has been reported that the economic crisis has had a negative impact on both physical and mental health in the general population which may account for its probable higher rate of psychiatric disorders in the low-income countries from Africa and the Middle East [27]. Our study shows that depression and anxiety are highly prevalent conditions in non-workers and in participants who do not have medical insurance. Some participants in our study are mainly from very low-income level without medical insurance, therefore they have no choice but to opt for more costly mental health services provided by private clinics. The strikingly high prevalence of depressive symptoms in our study is quite alarming because it suggests that locals with low incomes are at higher risk of depression, in accordance with findings from different countries [47,48].

The current results show that both anxiety and depression symptoms are higher with couples living among extended families. Such family conflict is comparable to those found in other studies [24,38]. Positive social interaction enables the creation of strong bonds and the development of a solid support network. Furthermore, personality can influence coping strategies and the ability to overcome adversity [24,38]. Jordan is a traditional society, and because of the remnants of its tribal origins, extended families still heavily influence social life positively by supporting family members and negatively by interfering with members’ social lives [38]. Palestinian couples in general and those who were living in an extended family type in specific had higher HAM-A and BDI-II scores compared to the Jordanian participants. This could be attributed to the persistent psychological pressure surrounding the political circumstances of oppression, lack of freedom, and many other occupational challenges [24]. The current study shows that around 19% of couples had suicidal thoughts and wishes and such a serious finding is comparable with other global studies which showed that anxiety and depression have significant effects on people’s quality of life, and are major risk factors for suicide [13,14]. In this study, poor to fair self-rated health is a strong independent factor associated with clinically significant depressive and anxiety symptoms. Around one third of participants rate their health evaluation to be excellent. The participants who have rated themselves with excellent health evaluation have the lowest BDI-II and HAM-A scores.

Consequently, these indeterminable cultural and environmental factors may have increased the likelihood of underrecognised and underreported depression and anxiety [22,23,33]. There are several potential reasons for the underreporting of depression and anxiety in the Arab countries. First, both Jordanian and Palestinian patients tend to deny mental disorders because of the associated social stigma, and it is considered a shame to be manifesting a mental disorder [49,50]. Hence, many Jordanian and Palestinian patients may shy away from seeking medical help. Even if they choose to consult their physicians, they may only complain of somatic symptoms, leading to incorrect diagnosis [49]. Moreover, underdiagnosis of psychological problems might be related to the participants’ education levels. More than one third of the study participants were educated beyond high school level. Postgraduate participants had the least BDI-II and HAM-A scores while illiterates had the highest scores. This could indicate that the higher the education levels, the better BDI-II and HAM-A scores. In the Middle and Far East, many studies have shown an association between poor education and poor mental health, which is partially mediated by health literacy [40,51]. This could suggest that many recruited participants with clinically significant depressive and anxiety symptoms did not seek medical help as they lack the knowledge about the clinical presentation of mental problems.

Addressing fertile couples’ depression and anxiety in Arabs entails taking into account all individual, societal, and environmental risk factors that could contribute to the escalation of depressive anxious symptoms, as should be the case in every culture.

A serious shortage of mental health professionals, such as psychiatrists, psychologists, and psychiatric nurses, is present in both Jordan and Palestine. Additionally, even though the nation has agreed in principle to incorporate mental health services into the primary healthcare system, implementation has been extremely restricted [52].

According to researchers, primary care practitioners need to educate fertile couples on mental disorders by inquiring about risk factors and significant symptoms related to functioning in the areas of cognition, social interaction, academic performance, and family functioning.

### 4.1. Limitations

Firstly, participants in the study were purposively selected which limits claims to typicality and generalisability. Hawthorne effect and over or under representation of measured outcomes is another limitation inherently associated with interview-style data collection. The relatively small sample of this descriptive study needs to be taken into consideration in future works. One last limitation is that the anxiety and depression details have not been evaluated in a standardised way or adequately, and in-depth interviews and multidimensional measurements may be required for this assessment.

### 4.2. Future Works

The current study establishes and compare a national prevalence of the occurrence and intensity of depression and anxiety symptoms among Jordanian and Palestinian fertile couples; as a result, it may offer a strong foundation from which to conduct future research to establish the most effective clinical approaches for educating, preventing and treating such mental problems. The second phase of this project will discuss the stress associated with infertility among infertile couples, and the last step will discuss the impact of COVID-19 pandemic on the psychological well-being of this target population (childbearing age group).

## 5. Conclusions

Our findings ought to alert healthcare professionals and other interested parties that there is a high burden of anxiety and depression symptoms in among Jordanian and Palestinian couples. To support Jordanian and Palestinian couples’ mental health, healthcare professionals, researchers, and educators should concentrate on creating efficient and culturally relevant preventive, and intervention procedures utilising evidence-based guidelines.

The lack of mental health services and the stigma attached to mental illness, on the one hand, and poverty, unemployment, and a lack of realistic hopes for a decent future in the context of regime corruption and perceived social injustice, on the other, exacerbate the stress of conflict and war. Therefore, poor mental health among fertile couples in Jordan and Palestine not only affects their immediate health but also has long-term detrimental repercussions on the health and well-being of the couples, affecting subsequent generations as well as society as a whole.

## Figures and Tables

**Table 1 healthcare-10-01679-t001:** Socio-demographic characteristics of the study sample (*N* = 469 *).

Parameter	Mean (SD)	*N* (%)
Country		
Jordan		285 (60.8)
Palestine		184 (39.2)
Gender		
Male		237 (50.5)
Female		232 (49.5)
Age in years	36.0 ± 9.70	
Body mass index (BMI)		
Male	26.7 ± 4.9 (overweight)	
Female	26.5 ± 5.3 (overweight)	
Smoking status		
Smoker		125 (26.7)
Non-Smoker		291 (62.2)
Ex-Smoker *		26 (5.6)
Pipe (Shisha)		26 (5.6)
Education level		
Illiterate		37 (7.9)
High school degree		134 (28.7)
BSc		190 (40.7)
Post graduate degree		106 (22.7)
Employment		
Yes		338 (72.5)
No		128 (27.5)
Family type		
Nuclear		244 (52.8)
Extended		218 (47.2)
Period of marriage in years	10.5 ± 7.9	
Insurance		
Yes		368 (78.6)
No		100 (21.4)
Insurance type		
MOH		241 (64.9)
Private		94 (25.3)
Others		36 (9.7)
Total monthly income in JOD	827 ± 655	
Suffering from chronic diseases (hypertension, diabetes, heart diseases, respiratory problems…etc)		
Yes		48 (10.8)
No		398 (89.2)
Your evaluation of your health		
Excellent		141 (31.5)
Very Good		221 (49.4)
Good		67 (15.0)
Accepted		18 (4.0)
Bad		0 (0.0)

*: Ex-Smoker: quit smoking for more than 6 months. JOD: Jordanian dinar, MOH: ministry of health. Note: Some data are missing, subsequently totals do not always add to 469.

**Table 2 healthcare-10-01679-t002:** Summary of BDI-II and HAM-A scores and grading (*N* = 469).

	Mean ± SD	Number of Participants and Percent
**Depression characterisation**		
Total BDI-II score	15.3 ± 10.0	
Normal (1–10 points)		160 (34.1)
Mild mood disturbance (11–16 points)		90 (19.2)
Borderline clinical depression (17–20 points)		66 (14.1)
Moderate depression (21–30 points)		120 (25.6)
Severe depression (31–40 points)		28 (6.0)
Extreme depression (>40 points)		5 (1.1)
Patients with normal BDI-II score and different HAM-A scores (Expected Anxiety alone)		
Mild severity (<17 points)		158 (98.8)
Mild to moderate severity (18–24 points)		2 (1.3)
Moderate to severe (25–30 points)		0 (0.0)
Very Severe (31–56 points)		0 (0.0)
Patients with suicidal thoughts and wishes		87 (18.6)
**Anxiety characterisation**		
Total HAM-A score	12.3 ± 8.2	
Mild severity (<17 points)		323 (68.9)
Mild to moderate severity (18–24 points)		114 (24.3)
Moderate to severe (25–30 points)		28 (6.0)
Very Severe (31–56 points)		3 (0.6)
Patients with normal HAM-A score and different BDI-II scores (Expected depression alone)		
Normal (1–10 points)		158 (48.9)
Mild mood disturbance (11–16 points)		62 (19.2)
Borderline clinical depression (17–20 points)		41 (12.7)
Moderate depression (21–30 points)		53 (16.4)
Severe depression (31–40 points)		8 (2.5)
Extreme depression (>40 points)		1 (0.3)

**Table 3 healthcare-10-01679-t003:** Correlation between BDI-II and HAM-A scores and different participants’ categorical demographic variables (*N* = 469).

Parameter	Total BDI-II Score		Total HAM-A Score	
Country				
Jordan	11.1 ± 10.1		8.4 ± 7.6	
Palestine	21.8 ± 5.4		18.3 ± 4.3	
	Jordan	Palestine	Jordan	Palestine
	*p* value		*p* value	
Gender	0.015	0.91	0.017	0.18
Smoking status	0.81	0.95	0.67	0.20
Family Type	0.04	0.90	0.011	0.21
Education level	0.001	0.57	0.048	0.89
Employment	<0.001	0.15	0.010	0.27
Insurance	0.192	0.104	0.91	0.32
Health evaluation	<0.001	0.95	<0.001	0.41

Note: Significance at *p*-value less than 0.05 level.

**Table 4 healthcare-10-01679-t004:** Assessment of factors affecting HAM-A and BDI-II scores among Jordanian fertile couples (*N* = 469).

Independent Variables	Dependent Variable			
	HAM-A Score		BDI-II Score	
	Standardised Coefficients Beta	*p*-Value	Standardised Coefficients Beta	*p*-Value
Gender	0.170	0.015	0.127	0.062
Family Type	0.141	0.024	0.112	0.066
Education Level	–0.125	0.047	–0.166	0.007
Employment	0.069	0.319	0.139	0.040
Evaluation of health	0.226	<0.001	0.250	<0.001

Note: Significance at *p*-value less than 0.05 level.

## Data Availability

The data presented in this study are available on request from the corresponding author.

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
