# Peer review of "Prevalence and Predictive Factors of Masked Depression and Anxiety among Jordanian and Palestinian Couples: A Cross-Sectional Study"

_healthcare, 2022, doi:10.3390/healthcare10091679_

Round 1

Reviewer 1 Report

The paper under review concerns an investigation of levels of depressive and anxiety symptoms among married couples in Jordan and Palestine. While I appreciate the dire need for more research on participants from non-Western countries, I do have a number of concerns.

The study design is described very briefly, basically only as a “convenience sample” of married couples with no indication where this convenience sample might have come from. The restriction to married couples and the high rate of highly-educated participants was puzzling, until the very end of the discussion section, where it is suddenly mentioned: “Future studies … of couples undergoing infertility treatment could involve enlarging the sample size and including a greater number of infertility clinics.” I can only assume, therefore, that this sample is a convenience sample of married couples presenting for treatment of infertility, a conclusion strengthened by the involvement of authors affiliated with obstetrics and gynaecology. This is very far from a representative population sample! The fact that this is a sample seeking infertility treatment should be mentioned in the title, the abstract, the introduction, the description of the study design, and the discussion. I am sure a study on depression and anxiety in such a sample can also be of interest, but it definitely needs to be made clear.

The title, abstract, and text of the manuscript also suggest that this paper is about “masked depression and anxiety”, which is generally understood as depression and/or anxiety “hiding” behind somatic symptoms or other symptoms. Since the authors do not assess anything that the depressive or anxiety symptoms are hiding behind, I do not see any reason to refer to “masked” depression and anxiety. There is also a lot of reference to “undertreated” and “underdiagnosed” depression and anxiety, and the results section even mentions that the participants scoring above 20 on the BDI-II “met the criteria for major depression”. Since the study is based on questionnaires, it cannot determine whether the criteria for a disorder are met. The levels of anxiety and depression are fairly high, but we do not know how many of these participants can be diagnosed with anything and we certainly cannot know how many should be treated.

In general, the paper tends to “over-reach”. To provide some examples: “Female gender being consistently linked with depression and anxiety would explain the differences in readiness to demand therapy, mechanisms of coping, permission to cry when depressed …” – I do not see how a higher prevalence of depression in women explains all of that. The final sentence of the conclusion (“Screening for depression and anxiety should be conducted regularly to provide full evaluation and relevant psychiatric treatment”) is also a huge over-reach, considering my points in the previous paragraph as well as the fact that there is very little evidence for the effectiveness of screening. I think a little more restraint would be a good idea.

References sometimes seem irrelevant or suboptimal (i.e. there are much better papers to cite). E.g. citing a paper on “Prognostic models for predicting relapse or recurrence of major depressive disorder in adults” as support for the high and increasing prevalence of anxiety and depression, citing “Anxiety disorders: a comprehensive review of pharmacotherapies” as support for the statement that anxiety is similarly impairing as depression.

Smaller points:

The sample size calculation is based on “10 participants per predictor”. This is not really an adequate way to calculate sample size (which should be based on anticipated effect size), but I appreciate that this cannot be changed anymore, since the study has already been conducted.

All predictors are assessed in a univariable fashion. It may be of some interest to conduct a multivariable analysis (based on linear regression) as well, since many of these predictors are likely to be correlated.

Table 1 and 2: it appears that participants with missing data are included in the denominator for the percentages. This makes them harder to interpret. It would be preferable to list the sample size for each parameter and remove the people with missing data from the calculation of the percentage.

Table 3: This table really needs some kind of effect size. My preference would be to indicate the mean (SD) of the BDI and HAM-A score for each “level” of the categorical variable (e.g. for Palestinian and Jordanian participants separately). It is currently impossible to assess the magnitude of the association.

Author Response

Dear Reviewer

Reviewer 2 Report

Deema Jaber et colleagues attempted to study the prevalence and predictive factors of masked depression and anxiety among Jordanian and Palestinian couples in a cross-sectional study. This is one of the very rare investigations dealing with mental health aspects in the traditional Arabian culture where – as the researchers point out – the topic is associated with social stigma and the majority of people experience mental illness as a shame. Nevertheless, Deema Jaber et colleagues were able to include a total of four-hundred and sixty-nine subjects (NOT couples!) were eligible for inclusion and agreed to participate in the study which resulted in a most favourable response rate of 91.9% participants. Remarkably, the majority of Jordanian and Palestinian patients included graded themselves to be mildly anxious and moderately depressed.

In total, the manuscript is well written, the state of the art is up to date. The authors applied relatively simple and non-elaborated statistical solutions which however serve sufficient for answering the research questions.

Minor points:

1.      A cross-sectional study was conducted to explore explored the and predictive factors of masked or underdiagnosed anxiety and depression among Jordanian and Palestinian couples. Four- hundred and sixty nine were eligible for inclusion and agreed to participate in the study.

What is meant by “Four- hundred and sixty nine were eligible” – the reader is directed to “couples” – however, in reality the authors mean participants (otherwise there should be a total of 938 cases).

2.      “predictive factors”

The term is in contradiction to the cross sectional design of the study. Consider: “related factors” or “associations with…”

3.      The majority of the most participants have graded themselves to be mildly anxious and moderately depressed

The “majority of most participants” is an unnecessary doubling which should be omitted – please give the empirical numbers in the abstract.

4.      The tables in the text body of the manuscript have an unprofessional appearance. Please improve.

5.      Current study showed that depression…(line 263)

Instead: “Our study shows that….” Please check all tempi in the text body – many of the findings of your own study should be presented in presence.

Author Response

Dear Reviewer
